# An Ultrametric Random Walk Model for Disease Spread Taking into Account Social Clustering of the Population

**DOI:** 10.3390/e22090931

**Published:** 2020-08-25

**Authors:** Andrei Khrennikov, Klaudia Oleschko

**Affiliations:** 1International Center for Mathematical Modeling in Physics and Cognitive Sciences, Linnaeus University, SE-351 95 Växjö, Sweden; 2Centro de Geociencias, Campus UNAM Juriquilla, Universidad Nacional Autonoma de Mexico (UNAM), Blvd. Juriquilla 3001, 76230 Queretaro, Mexico; olechko@unam.mx

**Keywords:** disease spread, herd immunity, hierarchy of social clusters, ultrametric spaces, trees, social barriers, linear growing barriers, energy landscapes, random walk on trees

## Abstract

We present a mathematical model of disease (say a virus) spread that takes into account the hierarchic structure of social clusters in a population. It describes the dependence of epidemic’s dynamics on the strength of barriers between clusters. These barriers are established by authorities as preventative measures; partially they are based on existing socio-economic conditions. We applied the theory of random walk on the energy landscapes represented by ultrametric spaces (having tree-like geometry). This is a part of statistical physics with applications to spin glasses and protein dynamics. To move from one social cluster (valley) to another, a virus (its carrier) should cross a social barrier between them. The magnitude of a barrier depends on the number of social hierarchy levels composing this barrier. Infection spreads rather easily inside a social cluster (say a working collective), but jumps to other clusters are constrained by social barriers. The model implies the power law, $1-t^{-a},$ for approaching herd immunity, where the parameter *a* is proportional to inverse of one-step barrier Δ. We consider linearly increasing barriers (with respect to hierarchy), i.e., the *m*-step barrier $\Delta_{m}=m\Delta .$ We also introduce a quantity characterizing the process of infection distribution from one level of social hierarchy to the nearest lower levels, spreading entropy E. The parameter *a* is proportional to E.

## 1. Introduction

In this paper we present a new mathematical model of disease (say a virus) spread that differs from the standard SIR-like models [1,2,3,4] (see also [5,6,7,8,9,10] for recent applications of these models to the covid-19 epidemic). Its main assumption is the heterogeneity of the population with respect to disease spread: splitting of population into social clusters with hierarchic tree-like structure. Mathematically our model is based on the theory of random walks on hierarchic tree-like structures, ultrametic spaces. We used the results of the well known paper [11] on random walk on *p*-adic trees, the simplest hierarchic structures endowed with ultrametric. (We recall that a *p*-adic tree, where p>1 is a natural number, is the tree with *p* branches leaving each vertex.)

The main outputs of the paper that can be interesting for epidemiologists are the graphs presented in Section 4: the power law for asymptotics of the probability $P_{I}\left(C,t\right)$ to become infected in social cluster *C* in a long run of an epidemic; so to say at the phase of approaching of some level of immunity:(1)$$P_{I}\left(C,t\right)∼t^{-a},$$
where *a* is a parameter depending on the magnitude of a social barrier (see (Equation 3)). Hence, we can model dependence of $P_{I}\left(C,t\right)$ on the strength of preventative anti-epidemic measures established by state authorities (see Figures 2 and 3).

We quantify herd immunity with probability
(2)PIm(C,t)=1−PI(C,t)∼(1−t−a).

This is a sort of “integral immunity”, a combination of innate and adaptive components. For small values of parameter a, this function increases very slowly; cf. with Britton’s analysis [8] demonstrating that induced herd immunity level for COVID-19 in Sweden is substantially slower than the classical herd immunity level.

As was pointed out, our basic assumption is that a virus spreads in a population having the structure of hierarchically coupled social clusters (see Appendix A on information about such clustering of infection during the COVID-19 epidemic). A social cluster can be a collective of some enterprise or a state department, say clerks of the community office of some town or the personnel of some hospital. Inside such a cluster people still have a relatively high degree of social connection (see preprint [12], Appendixes 1,2, for details). However, during an epidemic authorities erect sufficiently high barriers between clusters and people terminate many sorts of social contacts. Thus, we proceed with the following basic assumption.

**Assumption** **1.** 

*Disease spread is coupled to the hierarchic social cluster structure of population.*


Starting with this assumption, we design a mathematical model implying the following:

**Consequence** **1.** 

*The epidemic has a relatively slow decay and approach to herd immunity (for sufficiently high preventative barriers).*


The problem of approaching herd immunity is very important for diseases coming as repeating waves, the first wave, the second wave and so on. The herd immunity is mathematically formalized through the probability $P_{I}\left(C,t\right)$ (see Figure 2, Section 4).

To model cluster dynamics, we explore random walk on ultrametric spaces [11]. It was widely used in physics and microbiology; see [11,13,14,15,16,17,18] and references therein. Geometrically ultrametric spaces have the tree-like structure. Random walks on trees describe dynamics on energy landscapes [11,13,14,15,16,17,18]. There are given energy barriers $\Delta_{m}$ separating valleys; movement from one valley to another valley is constrained by necessity to jump over a barrier between them.

In our model a virus (or its carrier) walks randomly in socially clustered society. It starts just from a single social cluster (this assumption is used for mathematical simplicity); it spreads relatively easily inside any cluster, but to approach other social clusters it should “jump over social barriers”. For linearly increasing (with respect to hierarchy) barriers between clusters, the basic parameter *a* of the model (see (Equation 1)) has the form:(3)a=lnp/Δ.
Here Δ is the magnitude of the elementary barrier for hopping between nearest social levels. A higher social barrier Δ implies slower growth of herd immunity. (In the physics of spin glasses, this parameter has the form a=Tlogp/Δ, where *T* is temperature. In [12], we proceeded as in physics by introducing a social analog of temperature. However, the notion of social temperature needs further justification and in the present paper we preferred to proceed without it.) Quantity lnp can be interpreted as the entropy of the process of a virus spreading into subclusters of a cluster, spreading entropy; see (Equation 17). Thus, larger spreading entropy of the social cluster tree implies quicker herd immunity.

The configuration space of dynamics is the tree of social connections between people. In epidemic modeling, it is natural to assume the presence of the hierarchic structure in social clusters of people, by ranging basic social parameters coupled to infection (see Section 2 and Appendix A for examples). This representation of social types as vectors of hierarchically ordered social coordinates has been already used by the first author and his collaborators in a series of studies in cognition, psychology and sociology [19,20,21,22,23,24].

Just before submission of his preprint [12], the first author found the recent paper of Britton et al. [25] in which the role of population heterogeneity in the spread of COVID-19 was analyzed. We remark that Britton contributed a lot in mathematical modeling of COVID-19 spread in Sweden. His models [7,8] were explored by chief epidemiologist Anders Tegnell to justify the Swedish policy with respect to the epidemic—no lock-down given the expectation of rapidly approaching herd immunity. On the basis of Britton’s models, the Swedish State Health Authority predicted (at the end of April 2020) that herd immunity would be approached already in May. However, this prognoses did not match the real situation and herd immunity was not approached either in May or in June (see, e.g., [26,27,28] for reports from Public Health Institute of Sweden; see also [29,30,31,32,33]). In previous modeling [7,8] for the COVID-19 epidemic, the Swedish population was considered homogeneous. In [25], the heterogeneity of the population was considered as an important factor; the model involves two “social coordinates” (in our terminology, see Section 2): social activity and age.

Taking into account population social clustering is the basic similarity of our models (see also Appendix A) and generally paper [25] supports our approach. The main difference is that in [25] the hierarchic structure of social clustering and hence the hierarchy of barriers between clusters is not taken into account. Another crucial difference is in mathematical methods, based on the real metric vs. ultrametric. Surprisingly, these two totally different mathematical models led to graphs of the same shape; see Figure 3 (Section 4) and see Figure S2, Supplementary Material [25] (see also the remark after Figure 3.

Both models provide the possibility to play with the strengths of preventive measures and see their effects on the epidemics’ dynamics.

We do not want to overshadow our model of disease spread with mathematical technicalities. Therefore, we appealed to the simplest theory developed in [11], random walks of *p*-branching trees, where *p* is the same for all vertices. The general mathematical theory is based on theory of diffusion equations over the field of *p*-adic numbers $Q_{p}$; see, e.g., pioneer papers [34,35,36,37,38,39,40]. We remark that in contrast to the present paper the use of the *p*-adic diffusion equation is restricted by the constraint “*p* is a prime number”. The latter is crucial to defining the operation of division on a *p*-adic tree and determining the structure of the number field. Following [11], we proceed with an arbitrary natural branching number p>1. Of course, in real applications the branching number can depend on vertexes. The corresponding mathematical theory is more complicated [38,39].

Finally, we point out that recently ultrametric dynamical models for hierarchic clustering started to be used in geophysics, with applications to petroleum research; see, e.g., [41,42,43]: propagation of oil through capillary network in porous disordered media.

## 2. Social Trees

We represent human society as a system of hierarchically coupled clusters. Each cluster can be represented as a disjointed union of sub-clusters corresponding to the next level of hierarchy. Said population clustering can be done in many ways. We explore the approach that was used in a series of works of ultrametric modeling in cognition and sociology (see, e.g., [19,20,21,22,23]). The tree-like representation of social types is based on the selection of hierarchically ordered social factors enumerated as m=0,1,2,3,…; factor m=0 is the most important, m=1 is less important and so on. A social type is represented by a vector
(4)$$x=(x_{0},x_{1},\ldots ,x_{n-1}),$$
where its coordinates $x_{m}$ take (typically) discrete values quantifying the *m*-th factor. In the simplest case, $x_{m}$ takes two values, “yes” or “no”; 1 or 0. We call numbers $\left(x_{m}\right)$ social coordinates. The vector representation of social types and individuals is widely used in sociology and psychology. The main distinguishing feature of our model is endowing the space of vectors with the special metric reflecting the hierarchic structure corresponding to the order of social factors. The space of all vectors of the form (Equation 4) is called the hierarchic social space.

Since states play the crucial role as the epidemic-policy determining authorities, it is natural to select the most important index m=0 as a label for the states. However, we proceed with modeling the epidemic-situation in a fixed state, and influenced by lessons of COVID-19 epidemic, we use index m=0 for an individual’s age, and then m=1 for the presence of one chronic disease, m=2 for gender, m=3 for race, m=4 for a town, m=5 for a district, m=6 for profession, m=7 for the level of social activity, m=8 for the number of children and so on. We understand that said ranking of the basic social factors related to disease spread is incomplete (see Appendix A for further discussion). The contributions of sociologists, psychologists and epidemiologists can improve the present model essentially; see even the recent article [44] on the mathematical model of the evolutionary creation of social types and contribution of genetics and natural selection.

For mathematical simplicity, we consider *p*-adic coordinates, $x_{m}=0,1,\ldots ,p-1,$ where p>1 is a natural number. The space of all such vectors is denoted by the symbol $Z_{p;n}$ (*p* is fixed); *p*-adic social space. We now turn to the definition of a metric on $Z_{p;n}$ corresponding to hierarchy of coordinates. Consider two social vectors $x=(x_{1},\ldots ,x_{n-1})$ and $y=(y_{1},\ldots ,y_{n-1}).$ Let their first *k* coordinates be equal, $x_{0}=y_{0},\ldots ,x_{k-1}=y_{k-1},$ but the *k*th coordinates be different, $x_{k}\neq y_{k}.$ Then the hierarchic social distance between these social types should be d(x,y)=n−k. The first social coordinates are the most important: the common initial segment of vectors corresponds to a closer social sphere; an increase of *k* implies a decrease of distance between two social types. For example, let k=n−1; i.e., two points differ only by the last coordinate; then d(x,y)=1. This is the minimal possible distance in $Z_{p;n}.$ (The coordinate $x_{n}$ has the minimal degree of importance.) If the vectors differ already by the first coordinate, i.e., $x_{0}\neq y_{0},$ then d(x,y)=n. This is the maximal possible distance between points in space $Z_{p;n}.$ Distance *d* is ultrametric; it satisfies the strong triangle inequality:(5)d(x,y)≤max{d(x,z),d(y,z)},
for any triple of points $x,y,z\in Z_{p;n}.$ Here in each triangle, the third side is less than or equal not only to the sum of two other sides (as usual), but even to their maximum.

As usual, in a metric space we can introduce balls, $B_{N}\left(a\right)=\left\{x\in Z_{p;n}:d\left(a,x\right)\leq N\right\},$ where N=1,…,n, and $a=(a_{0},..,,a_{n-1})$ is some point in $Z_{p;n},$ the ball’s center. In an ultrametric space, any two balls are either disjointed or one is contained in another and any point of a ball can be selected as its center.

For our modeling, it is important that the space $Z_{p;n}$ can be split into disjoint social clusters. (As we shall see soon, these clusters are, in fact, balls.) Each cluster is determined by fixing the first few (the most important) social coordinates,
(6)$$Z_{p;n}=\cup_{j=0}^{p-1}C_{j},$$
where $C_{j}=\left\{x:x_{0}=j\right\}.$ This cluster representation corresponds to the first level of social hierarchy; we distinguish points by their most important coordinate. Each of clusters $C_{j}$ can be represented similarly as
(7)$$C_{j}=\cup_{j=0}^{p-1}C_{ji},$$
where $C_{ji}=\left\{x:x_{0}=j,x_{1}=i\right\}$ are clusters of the deeper hierarchic level and so on, up to the single-point clusters corresponding to fixing all social coordinates. Clusters are, in fact, ultrametric balls:(8)$$C_{i_{0}\ldots i_{k-1}}=\left\{x:x_{0}=i_{0},\ldots ,x_{k-1}=i_{k-1}\right\}=B_{n-k}\left(a\right),$$
where *a* is any point of the form $a_{0}=i_{0},\ldots ,a_{k-1}=i_{k-1}$ and arbitrary coordinates $a_{j},j=k,\ldots ,n-1.$

Geometrically space $Z_{p;n}$ is represented as a tree with *p* branches leaving each vertex; see Figure 1 for $Z_{2;3}.$ A cluster is a bunch of branches with a common root. By extending this root we split the cluster into sub-clusters. The vertexes of the ground level can be enumerated by natural numbers x=0,1,2,3,4,5,6,7. Distance between these numbers differs from the usual distance between natural numbers (the real line distance); we have: d(0,1)=d(2,3)=…=d(6,7)=1 (one step elevation to pass the barrier), d(0,2)=d(0,3)=d(1,2)=d(1,3)=2,d(4,6)=d(4,7)=d(5,6)=d(5,7)=2 (two steps elevation to pass the barrier), d(0,6)=…=d(3,7)=3 (three steps elevation to pass the barrier). This distance is associated with the hierarchic structure of the tree.

The social coordinate representation (Equation 4) is coupled with enumeration of vertexes of the ground level by natural numbers in the following way. Enumerate branches, leaving any vertex of the intermediate levels at 0 (left) and 1 (right). Each point of the ground level is represented by a vector—the compound branch going from the tree’s root to this vector:$$x=\left(x_{0}x_{1}x_{2}\right)=\left(000\right),\left(001\right),\left(010\right),\left(011\right),\left(100\right),\left(101\right),\left(110\right),\left(111\right).$$

This is listing the compound branches, i.e., going from the tree’s root through intermediate vertexes to the ground level; listing from the left-hand side to the right-hand side of the tree in Figure 1. This representation of tree’s compound branches is transferred into natural numbers as follows: $x=x_{2}+x_{1}2+x_{0}2^{2}.$ Let the above consideration related to epidemic $x_{0}=0,1$ (age: young, old), $x_{1}=0,1$ (chronic disease: absent, present), $x_{2}=0,1$ (gender: man, woman). Then, for example, 1 represents a young woman without chronic disease and 7—an old woman with chronic disease.

Now we consider the procedure of extension of a social tree by adding new social coordinates, so from tree $Z_{p;n}$ to tree $Z_{p;N},$ where N>n. As the result of such an extension, each point of social space $Z_{p;n}$ becomes a social cluster in social space $Z_{p;N}.$ In principle, it is impossible to determine a social type by fixing any finite number of social coordinates. Hence, we have to consider infinite sequences of coordinates:(9)$$x=\left(x_{0},x_{1},\ldots ,x_{n-1},\ldots \right),\;x_{j}=0,1,\ldots ,p-1.$$

Denote the space of such sequences by the symbol $Z_{p}.$ This is the complete hierarchic social space. Points of finite trees represent social clusters.

## 3. Probability to Become Infected from the Virus—Random Walk in a Hierarchic Tree of Social Clusters

Consider a tree-like structured population with *n* levels of social hierarchy. Mathematically this structure is described by social space $Z_{p;n}$ endowed with ultrametric d. Balls determine the social cluster partitions of $Z_{p;n}$; see (Equation 8) and (Equation 6), (Equation 7).

The fundamental quantity of our modeling of epidemic is the probability to become infected at the instant of time *t* for a person belonging to a social cluster *C* (some ball in social space). Denote this probability by the symbol $P_{I}\left(C,t\right).$ We are interested in its dynamics and more precisely in its asymptotic behavior for large t. This stage of epidemic can be considered as the stage of approaching herd immunity.

Now we present the interpretation of this probability in terms of the virus’s random walking in a population that is tree-like clustered. A virus plays the role of a system moving through barriers in models of dynamics on energy landscapes (see [11,13,14,15,16,17,18] and references herein). In our case, these are social barriers between social clusters of the population. The virus performs a complex random walk motion inside each social cluster moving in its sub-clusters, goes out of it and spreads through the whole population; sometimes the virus comes back to the original cluster from other social clusters that have been infected from this initial source of infection, and so on. During this motion the virus should cross numerous social barriers. Denote by P(C,t) the probability to find a virus in social cluster C. This probability is interpreted as in statistical mechanics of gases: as the concentration of virions (virus particles, consisting of nucleic acid surrounded by a protective coat of protein called a capsid) in cluster C. Now, we identify probabilities, $P\left(C,t\right)=P_{I}\left(C,t\right):$ probability to become infected is determined by concentration of virions in this cluster. Of course, concentration of virions is coupled with concentration of infected people, but not straightforwardly, since

Virions can live on various surfaces;The COVID-19 epidemic demonstrated the crucial role of superspreaders—super-powerful sources COVID-19 virions [45] (see Appendix B).

We do not want to go into detail, since the dynamics of the probability P(C,t) were well studied in physics and microbiology; see [11,13,14,15,16,17,18] and references herein. The asymptotics for t→∞ (relaxation regime) depend crucially on the barriers’ magnitude and how rapidly they grow up on the way from one cluster to another.

## 4. Dynamics of the Probability to Become Infected

As in the previous section, we consider a random walk on a finite tree. Here we follow the paper of Ogielski [11]. Let us consider a finite tree with *n* levels. Thus, there are $p^{n}$ points at the last level. They enumerate the total population: $x=0,\ldots ,p^{n}-1.$

Let a virus encounter a barrier of size $\Delta_{m},$ in hopping a distance *m* (crossing *m* levels of hierarchy), where $\Delta_{1}<\Delta_{2}<\ldots <\Delta_{m}<\ldots .$ It is supposed that barriers $\Delta_{m}$ are the same for all social clusters, i.e., they depend only on distance, but not on clusters.

Consider the tree at Figure 1. We identify the lengths of branches between vertexes with magnitudes of barriers. Then the barriers on this tree depend on clusters, so from this viewpoint the social tree is not homogeneous.

The probability to jump over the barrier $\Delta_{m}$ has the form (up to the normalization constant):(10)$$P\left(m\right)=e^{-\Delta_{m}}.$$

The meaning of this formula is straightforward: probability to jump over a higher barrier is smaller.

Consider the energy landscape with a uniform barrier Δ, at every branch point; that is, a jump of distance 1 involves surmounting a barrier Δ, of distance 2, a barrier 2Δ and so on. Hence, barriers linearly grow with distance m,
(11)$$\Delta_{m}=m\Delta ,m=1,2,\ldots .$$

Barriers $\Delta_{m}$ are sufficiently high, but they still are not walls of the lock-down type. The probability to jump over the barrier $\Delta_{m}$ has the form (up to the normalization constant):(12)$$P\left(m\right)=e^{-m\Delta }.$$

In particular, the probability of jumping to the nearest clusters equals $R=e^{-\Delta }.$ It exponentially decreases with increase of barrier Δ.

The power law asymptotics given by formula (Equation 15), see below, are obtained by solving of the master equation for random walk on the *p*-adic tree. For finite *n* (the number of levels of the tree $Z_{p;n}),$ this equation can be solved exactly (this is the main result of [11]). Consider the initial condition P(x,0)=δ(x); we recall that the points of $Z_{p;n}$ can be represented by natural numbers $x=0,\ldots ,p^{n}-1.$ The solution with this initial condition has the form:(13)$$P_{I}\left(x,t\right)=\frac{1}{p^{n}}+\frac{1}{p}\sum_{m=0}^{n-1}\exp\left\{-m\ln p-\frac{1}{1-R}\left[\left(p-R\right)R^{m+1}-R^{n+1}\right]t\right\},$$
where $R=e^{-\Delta }$ is the probability of jumping to the nearest clusters. Then one sends n→∞ and uses that R<1; finally the asymptotic law (Equation 15) for t→∞ can be found. The same asymptotics can be derived for any initial condition of the form P(x,0)=δ(x−y), where *y* is some fixed point of the configuration space $Z_{p;n}.$

By using random walk on the tree with *n* levels of hierarchy and approaching n→∞, one can derive the following asymptotic behavior of the probability: P(x,t) [11], and hence, the probability to become infected $P_{I}\left(x,t\right)$; in our model, the latter is equal to the former:(14)$$P_{I}\left(x,t\right)=P\left(x,t\right)∼t^{-\ln p/\Delta },t\to \infty .$$

Since any social cluster *C* is given by an ultrametric ball and a ball can be represented as union of *x* points; the same asymptote can be derived for any social cluster (ball) C:(15)$$P_{I}\left(C,t\right)=P\left(C,t\right)∼t^{-\ln p/\Delta },t\to \infty .$$

Set a=lnp/Δ. If a≪1, i.e., the primary social barrier Δ is relatively large, then the probability for a person in the social cluster *C* to become infected decreases rather slowly; see Figure 2.

Hence, immunity increases also slowly (see Figure 3), as function
(16)PIm(C,t)≡1−PI(x,t)∼1−t−lnp/Δ,t→∞.

Thus, for the low preventive level Δ=B, herd immunity increases sufficiently quickly; increasing the preventive level makes the growth of herd immunity essentially slower. We remark that these graphs have the same shape as graphs (for COVID-19 epidemic) obtained in the recent paper [25]; see Figure S2, for age and activity structured community (besides the initial segment of dynamics when the number of infected is very small; but we recall that our model provides the asymptotic behavior, so it does not describe the initial phase of epidemic).

The same asymptotics can be obtained in terms of *p*-adic diffusion; see [40]. However, the latter theory is more complicated mathematically; see also [38,39] for diffusion on general ultrametric spaces represented geometrically by arbitrary trees.

Finally, we note that parameter *a* also depends on *p* the branching index of the social tree. We recall that the mathematical model of this tree is idealized; the branching index is constant—it does not depend on a vertex. Each cluster determined by *k* social coordinates, $C_{i_{0}\ldots i_{k-1}}=\left\{x:x_{0}=i_{0},\ldots ,x_{k-1}=i_{k-1}\right\},$ can be split into *p* subclusters $C_{i_{0}\ldots i_{k-1}i}=\left\{x:x_{0}=i_{0},\ldots ,x_{k-1}=i_{k-1},x_{k}=i\right\},i=0,1,\ldots ,p-1.$ The parameter *p* determines the complexity of the social clustering of a population. By Equation (Equation 15), an increase of *p* implies a speed up in decreasing the probability $P_{I}\left(C,t\right)$ or in other words a speed up of increasing the herd immunity. For the same one-step social barrier Δ, herd immunity is approached quicker in a population with a complex structure of social relations, large parameter p. The slowest dynamics correspond to the p=2: “yes–no” system of social coordinates; say there are just two districts in a town, one populated by people with high income and another by people with low income. The quantity lnp can be interpreted statistically as entropy of the process of distribution of infection into *p* subclusters coupled to a vertex. Suppose that a virus can spread with equal probability $q_{i}=1/p$ into each of the subclusters $C_{i_{0}\ldots i_{k-1}i}$ of the cluster $C_{i_{0}\ldots i_{k-1}}.$ Entropy of this spreading equals to
(17)$$E=-\sum_{i=0}^{p-1}q_{i}\ln q_{i}=\ln p.$$

In terms of spreading entropy, asymptotics (Equation 16) can be rewritten as
(18)PIm(C,t)∼1−t−E/Δ,t→∞.

Thus, larger spreading entropy of the social cluster tree implies quicker approaching herd immunity.

Our conjecture is that this formula is valid for more general process of infection spread, with nonuniform distribution for probabilities $q_{i}.$

## 5. Average Social Distance Traveled by Disease Spreader

In our mathematical model, when any disease spreader travels through the social tree, he/she visits a few social clusters and infects people in these clusters. The theory of random walks in ultrametric spaces predicts the average social distance for spreader’s travel through clusters, starting at some fixed cluster *x* and jumping to other cluster y,
(19)$$\langle d\left(x,t\right)\rangle =\sum_{y\in Z_{n;p}}d\left(x,y\right)P\left(y,t\right)$$

For linearly growing social barriers, and n→∞, the asymptotic behavior has the following form:(20)$$\langle d\left(x,t\right)\rangle ∼\frac{\log t}{\Delta },\;t\to \infty .$$

This result can be derived by a scaling argument: If the time is rescaled by a factor *R* (where $R=e^{-\Delta }$, then all sets of neighboring points on the lowest level of tree $Z_{n;p}$ become indistinguishable, and we are left with an effective lattice which is one level lower. This results in a shift of 1 in the ultrametric which leads to asymptotics (Equation 20).

This average distance goes to infinity. As can be expected, a lower one-step social barrier Δ induces more rapid growth. Although the log-growth is relatively slow, it, nevertheless, implies very extended spread of the infection. Unboundness of 〈d(x,t)〉 can be associated with the presence of super-spreaders who jump even over high social barriers and spread the virus to social clusters that are far from the original source of infection. We repeat once again that the distance under consideration is in social and not in physical space.

## 6. Concluding Remarks

The presented ultrametric model with random walk dynamics on energy landscapes describes disease spread in the socially clustered population. This approach provides the possibility to account for the dependence of the epidemic’s dynamics on the strength of barriers between social clusters. Graphs in Figure 2 and Figure 3 show the differences between the epidemic’s dynamics for relatively mild and strong preventative measures. Such measures inhibit approaching herd immunity; higher barriers imply stronger inhibition. Generally even mild preventative policy approaches herd immunity with asymptotics given by the power law. The model elevates the role of the social dimension of disease spreading compared with its purely bio-medical dimension (see also Appendix A).

We applied to the new area of research, to epidemiology, mathematical theory that was developed for applications in statistical physics (spin glasses) and microbiology (protein folding): ultrametric random walk describing dynamics on complex energy landscapes with the hierarchic structure of barriers between valleys. The presence of social barriers growing with hierarchy’s levels makes the evolution of epidemic essentially slower than in models which do not take into account the cluster-character of infection spreading.

As was mentioned, we used the very simple mathematical model for random walk on ultrametric space (having hierarchic tree-like structure. Mathematical development of this approach can go in two directions: (a) using *p*-adic analysis and diffusion [34,35,36,37,38,39,40], especially the approach developed in paper [40]; (b) considering more complex models of social networks and lattice dynamics [46,47,48,49,50].

Although the model is very simplified, it reflects the basic features of disease spread in the socially clustered population. We hope that our model will stimulate further development of ultrametric epidemiological models.

## Figures and Tables

**Figure 1 entropy-22-00931-f001:**
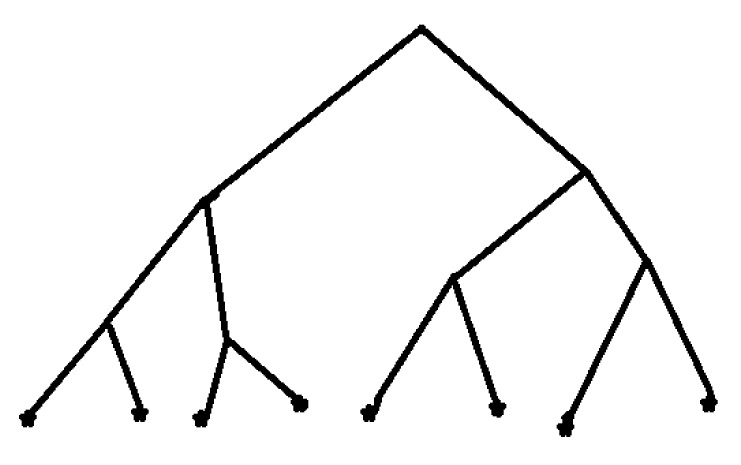
Mean reflectance spectra of locally acquired and self-grounded red chili.

**Figure 2 entropy-22-00931-f002:**
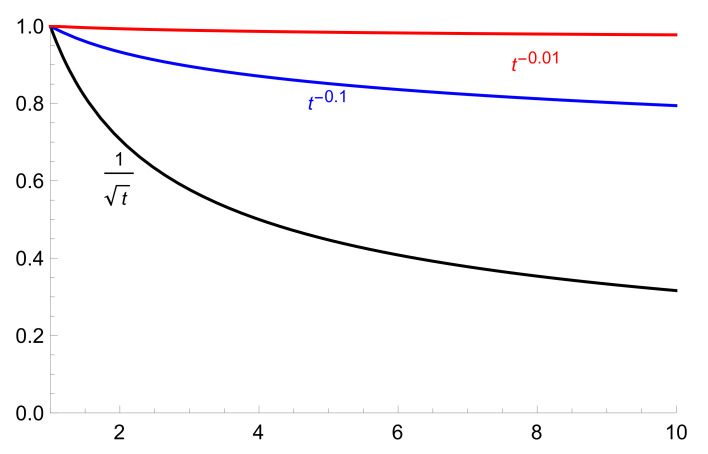
Asymptotic behavior of probability to become infected, Δ=B,5B,50B, where B=2logp (lowest, middle*, and upper graphs).

**Figure 3 entropy-22-00931-f003:**
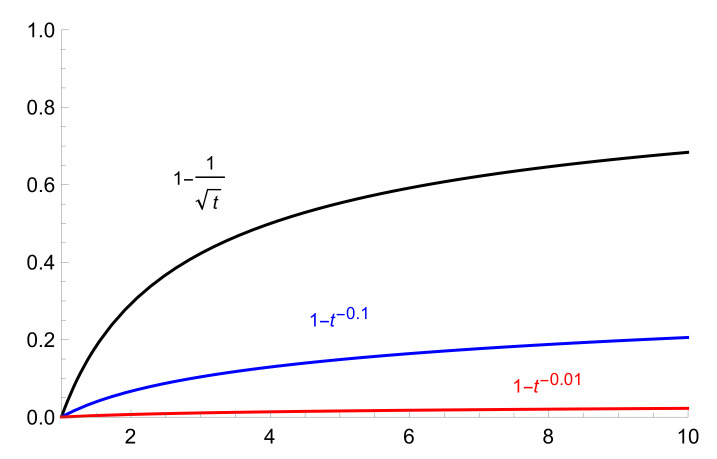
Asymptotic behavior of probability to become immune, Δ=B,5B,50B, where B=2logp (upper, middle and lowest graphs).

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
