# Peer review of "An Ultrametric Random Walk Model for Disease Spread Taking into Account Social Clustering of the Population"

_entropy, 2020, doi:10.3390/e22090931_

Round 1

Author Response

I would like to thank the reviewer for careful reading of the paper and numerous corrections of the style and misprints, and English!

Reviewer 2 Report

See report.

Author Response

I would like to thank the reviewer for positive evaluation of my paper and its careful editing.

Reviewer 3 Report

The authors apply the theory of random walk on the energy landscapes represented by ultrametric spaces (having tree-like geometry) to build a mathematical model of disease spread taking into account the hierarchic structure of social clusters in population. It describes dependence of epidemic’s dynamics on strength of barriers between clusters which are established by authorities as preventing measures.

This work can be improved if real data are considered and the figures presented are discussed, On the other hand, Figure 1 does not appear.

Author Response

I would like to thank the reviewer for careful reading of the paper. Yes, figure 1 disappeared (in some way) from the revision. Now, I solved this problem. The paper is theoretical, but we plan to write another paper with data analysis (the second author  started to work on this, but it will take time). 

Reviewer 4 Report

I consider that the paper is well presented and well structured now. I recommend its publication on Entropy after having gone through it.

Author Response

Thanks a lot for positive evaluation of the paper.

This manuscript is a resubmission of an earlier submission. The following is a list of the peer review reports and author responses from that submission.

Round 1

Reviewer 2 Report

See attached file, Remarks 1 and 2, together with minor comments.

Reviewer 3 Report

This work is a direct application of theory of random walks in ultrametric spaces to explain slower approaching the herd immunity during the covid-19 epidemy in Sweden. Since this was predicted by a variety of other models, for its publication, it must be improved comparing  the results of the model with real data and with other works that have been recently published such as Britton-2020. Moreover, the abstract is too long can be written shorter.

The manuscript requires a revision to improve English and eliminate some fat-finger errors.

For example:

In line 29 and 182  says “ultarmetric spaces”.

In line 39 says “is is”

In line 73 says Fig. ??

In footer 4 of page 2 says “can hope over barriers”.

In line 156 says “on on”

In line 157 says “we a random walk”

In footer 5 of page 6 appears ??.

Reviewer 4 Report

This is a paper with an interesting idea: exploiting the results for thee ultrametric random walk model to explain slow approach to herd immunity in a social model described as an ultrametric space. The idea is plausible but the paper has several flaws that I consider impede the publication on Entropy. I will mention some of them in the following.

The title suggests that some specific features of covid-19 will be used in the model to be presented but the only specific mention to it in the construction of the model comes after introducing the energy barriers of the landscape (equation (6)). The author claims that a linear growth of barriers with distance seems to be the "most natural from the viewpoint of social connections during the covid-19 epidemy in Sweden" , however this assumption requires a stronger support and I don't find it in the paper. Same applies for "Sweden"; folllowing what the title suggests it seems that the Sweden case would be analyzed or extensively used in the construction of the model but this is not the case.

I don't think it is good for the scientific debate to mention a paper like [15] which as I understood tackles the same problem but then saying that a more careful study by the author is required. This has to be done before submission.

As far as I understood the author uses known mathematical results which came mainly from [19] and interpret them in the framework of his epidemiological model. This is by no means wrong but more stress on the meaning of the parameters is required to convince the reader that indeed these results have something to do with epidemics. For instance, I don't find in the paper any good explanation/discussion on the "social temperature".

Figures 2 and 3 display a very similar information . It does not make sense to present them separately.

Unfortunately, for the above presented reasons I don't think the paper can be accepted for publication on Entropy.